# Development of Mucoadhesive Electrospun Scaffolds for Intravaginal Delivery of *Lactobacilli* spp., a Tenside, and Metronidazole for the Management of Bacterial Vaginosis

**DOI:** 10.3390/pharmaceutics15041263

**Published:** 2023-04-18

**Authors:** Margaret O. Ilomuanya, Peace O. Bassey, Deborah A. Ogundemuren, Uloma N. Ubani-Ukoma, Alkiviadis Tsamis, Yuwei Fan, Konstantinos Michalakis, Pavimol Angsantikul, Abdulrahman Usman, Andrew N. Amenaghawon

**Affiliations:** 1Department of Pharmaceutics and Pharmaceutical Technology, Faculty of Pharmacy, University of Lagos, Lagos 100213, Nigeria; ofonabasi63@gmail.com (P.O.B.); ogundemurendeborah@gmail.com (D.A.O.); uubani-ukoma@unilag.edu.ng (U.N.U.-U.); 2Department of Mechanical Engineering, School of Engineering, University of Western Macedonia, 50100 Kozani, Greece; atsamis@uowm.gr; 3School of Engineering, College of Science and Engineering, University of Leicester, Leicester LE1 7RH, UK; 4Department of Restorative Sciences & Biomaterials, Henry M. Goldman School of Dental Medicine, Boston University, Boston, MA 02118, USA; 5Centre for Biomedical Research, Population Council, New York, NY 10065, USA; pangsantikul@popcouncil.org; 6Department of Biotechnology and Pharmaceutical Microbiology, Faculty of Pharmacy, University of Lagos, Lagos 100213, Nigeria; 7Department of Chemical Engineering, Faculty of Engineering, University of Benin, Benin City 300287, Nigeria; andrew.amenaghawon@uniben.edu

**Keywords:** bacterial vaginosis, electrospun scaffolds, *Lactobacilli* spp., mucoadhesion, polyvinyl alcohol, polycaprolactone

## Abstract

Bacterial vaginosis (BV) is an infection of the vagina associated with thriving anaerobes, such as *Gardnerella vaginitis* and other associated pathogens. These pathogens form a biofilm responsible for the recurrence of infection after antibiotic therapy. The aim of this study was to develop a novel mucoadhesive polyvinyl alcohol and polycaprolactone electrospun nanofibrous scaffolds for vaginal delivery, incorporating metronidazole, a tenside, and *Lactobacilli*. This approach to drug delivery sought to combine an antibiotic for bacterial clearance, a tenside biofilm disruptor, and a lactic acid producer to restore healthy vaginal flora and prevent the recurrence of bacterial vaginosis. F7 and F8 had the least ductility at 29.25% and 28.39%, respectively, and this could be attributed to the clustering of particles that prevented the mobility of the crazes. F2 had the highest at 93.83% due to the addition of a surfactant that increased the affinity of the components. The scaffolds exhibited mucoadhesion between 31.54 ± 0.83% and 57.86 ± 0.95%, where an increased sodium cocoamphoacetate concentration led to increased mucoadhesion. F6 showed the highest mucoadhesion at 57.86 ± 0.95%, as compared to 42.67 ± 1.22% and 50.89 ± 1.01% for the F8 and F7 scaffolds, respectively. The release of metronidazole via a non-Fickian diffusion-release mechanism indicated both swelling and diffusion. The anomalous transport within the drug-release profile pointed to a drug-discharge mechanism that combined both diffusion and erosion. The viability studies showed a growth of *Lactobacilli fermentum* in both the polymer blend and the nanofiber formulation that was retained post-storage at 25 °C for 30 days. The developed electrospun scaffolds for the intravaginal delivery of *Lactobacilli* spp., along with a tenside and metronidazole for the management of bacterial vaginosis, provide a novel tool for the treatment and management of recurrent vaginal infection.

## 1. Introduction

Vaginal biofilms play a role in bacterial vaginosis (BV) pathogenesis, resulting in the recurrence of infection [1]. The structure of the biofilm limits the antibiotic penetration into the matrix as well as the interaction with the bacteria [2,3,4]. These biofilms are dense structures that adhere strongly to vagina epithelial tissues and are composed of clusters of *Gardnerella vaginalis*, which aid the growth of anaerobic microorganisms [5,6,7,8]. There has been a high incidence of recurrence of BV infection in individuals who have undergone antimicrobial therapy for the condition. Therefore, drug delivery systems are needed that maximize therapeutic outcomes by penetrating the bacteria biofilm and, in the process, restore healthy vaginal flora via enhancing the growth of *Lactobacilli* spp. Vaginal infections require an effective concentration of antimicrobial drug at the infection site for complete disease eradication. Conventional vaginal treatments for BV, such as gel and creams, have often been linked to low retention time in the vagina and discomfort due to leakage, which significantly reduced patient compliance [9,10]. 

Electrospun nano-scaffolds are a novel and effective way to administer drugs vaginally. Nanofibers have exhibited adequate flexibility, high loading capacity, high mucoadhesive strength, and characteristic softness, making them suitable for vaginal drug application [11,12,13]. The vagina was enriched with mucins that act as attachment sites and aid the retention of the microbes [8]. It was imperative that the mucoadhesive nanofibers had prolonged residence time with no drug leakage [9] to optimize the release of the loaded API agent. In a study incorporating metronidazole into nanofibers with polyvinylpyrrolidone (PVP) for treatment of BV, it was observed that the mucoadhesive nature of the nanofibers embedded with the antimicrobial was increased with an increasing concentration of PVP due to the hydrogen bond between PVP and mucin, which led to prolonged residence in the vagina [10,11]. The utilization of a polymer blend in this study facilitated the incorporation of other active components, increasing its viability as a treatment alternative, as compared to previous studies utilizing only one polymer blend.

Reoccurrence and antimicrobial resistance were experienced by people diagnosed with bacterial vaginosis due to the increasing rates of treatment failure [1,9]. It is important that drug delivery systems can penetrate the bacteria biofilm and, in the process, restore healthy vaginal flora. The introduction of the amphoteric tenside sodium cocoamphoacetate was intended to reduce the surface tension between the biofilms and the scaffolds in order to completely clear the films and prevent recurrence. These electrospun scaffolds comprised polyvinyl alcohol and polycaprolactone and incorporated metronidazole and *Lactobacilli* spp., and they showed various advantages over other delivery systems in the treatment of BV, such as cost-effectiveness, ease of operation, and controlled release, which increased the contact time of the drug at the site of action (the vagina). Studies have elucidated that probiotic bacterium facilitated the modulation of cervicovaginal mucosal immune responses, followed by the inhibition of biofilm formation [10]. *Lactobacilli*-containing vaginal probiotics as BV interventions have been widely used with varying degrees of success, because for a probiotic regimen to be effective, a high number of beneficial *Lactobacilli* spp. must be loaded. This must be accompanied by and retain the viability and functionality of the bacteria until used. 

The aim of this study was to develop novel mucoadhesive, electrospun nanofibrous scaffolds for the vaginal delivery of a triad consisting of metronidazole, cocoamphoacetate as a tenside, and *Lactobacilli* spp. for the management of BV. This approach to drug delivery sought to combine an antibiotic for bacterial clearance, a tenside biofilm disruptor, and a lactic acid producer to restore the healthy vaginal flora, prevent BV recurrence, and ensure an improved therapeutic outcome. 

## 2. Materials and Methods

Polycaprolactone (MW: 50,000–80,000 CAS NO 24980-41-4), polyvinyl alcohol ((MW: 30,000–70,000 CAS NO 9002-89-5) Shanghai Macklin Biochemical Co. (Shanghai, China)), sodium cocoamphoacetate (CAS NO 68411-57-4), metronidazole B.P., (Sigma–Aldrich, St. Louis, MO, USA), *Lactobacillus* spp., (human vaginal epithelial cells) pH 4.5, and a phosphate buffer, prepared according to USP XXVII (Shanghai Macklin Biochemical Co., Ltd.) were obtained. Acetic acid ((MW: 60.05 CAS NO 64-19-7) Honeywell Fluka, Germany), chloroform ((MW: 119.38 g/mol. CAS NO 67-66-3), Merck KGaA 64271 Darmstadt Germany), ethanol ((46.07 CAS NO 64-17-5), Fisher Scientific UK (Loughborough, UK)), deionized water, Mueller–Hinton agar ((MHA), Merck KGaA, Darmstadt, Seelze, Germany), anaerobic gas packs ((Thermo Scientific, Leicestershire, UK)), normal saline B.P., and dextrose saline B.P. (Fidson Healthcare Plc, Ogun, Nigeria) were also obtained. All chemicals were of analytical grade and were used as obtained from suppliers without further purification.

### 2.1. Fabrication of Metronidazole/Sodium Cocoamphoacetate Loaded Nanofibres

A solution of PCL (20% *w*/*v*) was prepared by dissolving the polymer in a chloroform and ethanol solvent system (75:25) at 50 °C under magnetic stirring until the dissolution was complete. This solution was then left overnight to achieve complete dissolution before incorporating the polymer blend [14]. PVA solution (23% *w*/*v*) was prepared by dissolving in a deionized water/acetic-acid solvent system. The mixture was then transferred into a 200 mL conical flask and stirred magnetically at 300 rpm at 80 °C for 2 h and then at 50 °C for 1 h [15]. The PCL/PVA blend was prepared by adding the PVA solution to the PCL solution and then stirring for 4 h to ensure complete dissolution [14]. Metronidazole was added to the PCL/PVA solution at a concentration of 0.5% and magnetically stirred for 1 h for complete dissolution [10]. The amphoteric tenside, cocoamphoacetate, was added to the mixture at a concentration 0.5% *w*/*v* The final mixture was magnetically stirred at 300 rpm at 80 °C for 2 h to ensure complete dissolution. The solutions were stored for 24 h to ensure homogeneity before electrospinning was carried out using a Nanolab Instruments NLI Basic (model: PS35-PCL), Selangor, Malaysia [15].

### 2.2. Fabrication of Lactobacillus spp. Loaded Nanofibres

A stock culture of *Lactobacilli* spp. of two human Lactobacillus strains isolated from human vaginal tract identified via the online BLAST search with corresponding percentage identity and GenBank accession numbers were Lactobacillus fermentum (98.85% identity, SUB7940628 Seq_29B_BSF-8_ZTJ MT904660), Lactobacillus fermentum (98.66% identity, SUB7940628 Seq_30B_BSF-8_ZTJ MT904661) was sub-cultured in an MRS agar, which had been freshly prepared under sterile conditions. A total of 70 g of the MRS agar water powder was added to dextrose saline (1 L), and the solution was heated and stirred appropriately before being autoclaved at 121 °C. The bacteria were then harvested from the agar culture with 5 mL of a 1:2 skimmed milk/normal saline suspension and then added to the polymer mix of PCL/PVA. The mixture formed a white emulsion that was then stirred at 600 rpm at room temperature before electrospinning [16].

### 2.3. Electrospinning Process

The process of dual electrospinning was carried out under optimal conditions at a temperature of 25 ± 1.56 °C, a relative humidity (RH) at 40%, and a feed rate of 1.0 mL/h. Each electrospinning, as shown in Table 1, was loaded into 2 10 mL plastic syringes with an internal diameter of 0.8 mm and a length of 20 mm, and fitted with gauge 18G needles. The solutions were fed at a speed of 1.0 mL/h. The collector was placed 15 cm from the needle tip. Electrospun fiber scaffolds were collected with an applied voltage of 25 KV in the needle. The electrospun scaffolds were stored in airtight plastic bags [15]. 

### 2.4. Morphological Characterization of Fabricated Electrospun Scaffolds 

#### 2.4.1. Scanning Electron Microscopy 

The structural characteristics of the electrospun scaffolds were determined in triplicate using a Hitachi 6600 field emission scanning electron microscopy (Tarrytown, NY, USA). Backscattering images were acquired under variable pressure modes, at a 60 Pa vacuum with N_2_, and a 15 kV acceleration voltage. The scaffolds were cut into small pieces of 5 mm × 5 mm and mounted with carbon conductive tape on SEM stub. Each sample fiber was sputtered coated with a small layer of carbon before observation [15].

#### 2.4.2. Determination of the Porosity of the Scaffolds

The porosity of the electrospun scaffolds were assessed using the liquid displacement method with ethanol utilized as the displacement liquid (Equation (1)).
Porosity of the scaffolds = (V1 − V2)/(V2 − V3) (1)
where V1 = volume of ethanol used, V2 = volume of ethanol present in the ethanol-embedded scaffold after 5 min, and V3 represents the amount of ethanol left over after the ethanol-embedded scaffold was removed. The experiment was carried out three times [15]. 

### 2.5. Thermal Analysis of Scaffolds

A thermogravimetric analyzer (Shimadzu, DSC-60, Kyoto, Japan) was used to determine thermal stability of the scaffolds via differential thermal analysis (DTA) and thermal gravimetric analysis (TGA). A total of 2 mg samples were placed in aluminum pans. The samples were then heated in platinum pans from 25 °C to 200 °C at the heating rate 10 °C/min under nitrogen atmosphere [10].

### 2.6. Chemical Characterization

#### 2.6.1. Differential Scanning Calorimetry

Using differential scanning calorimetry, possible polymer interactions and temperature transitions were investigated and recorded (DSC: Mettler Toledo^®^, Columbus, OH, USA). Under nitrogen, scans were taken at temperatures ranging from 37 to 45 °C at a heating rate of 10 °C per minute [17]. 

#### 2.6.2. Attenuated Total Reflectance Fourier Transform Infrared Spectroscopy

To characterize the absorption bands of the scaffolds, spectra were recorded in the region of wave numbers between 500 and 4000 cm^−1^. Vacuum drying (45 °C) was used to dehydrate electrospun scaffold samples that were then placed atop a diamond crystal for FTIR analysis. Smoothing was performed where appropriate to reduce the noise, without the loss of any peaks. The absorption peaks/bands were investigated to determine if there was any chemical interaction occurring because of the presence of varying polymers and excipients [17]. 

#### 2.6.3. X-ray Diffraction 

Using a D8 diffractometer (Bruker, Billerica, MA, USA) in the CuK monochromatic radiation, the range employed was 10–80°, with a 0.05° step size for X-ray diffraction analysis. The specimens were heated up to temperatures of 600, 1000, 1200, and 1480 °C and then quenched. They were studied to identify phases after heat treatment and associate possible phase transformations with the observed thermal effects [18].

### 2.7. Mechanical Characterization of the Electrospun Scaffolds

#### 2.7.1. Ultimate Tensile Strength of the Electrospun Scaffolds

The electrospun scaffold’s tensile strength and ductility was determined using a Universal Testing Machine (Instron-series 3369^®^, Norwood, MA, USA) equipped with a 50 kN load cell. A material’s maximum fracture resistance was measured by its ultimate tensile strength (UTS). When the load was applied as simple tension, it was comparable to the greatest load that could be supported by a square inch of cross-sectional area. The ability of a material to be drawn or plastically deformed without breaking was assessed by ductility. The scaffolds were cut into 50 mm × 10 mm parts and tested at a temperature of 20 °C and a humidity of 60%. The scaffolds were then clamped together and stretched at a rate of 50 mm/min with a load range of roughly 50 N and a gauge length of 50 mm. At room temperature, the tensile strength was measured three times, and an average value was determined [15].

#### 2.7.2. Mucoadhesion In Vitro Analysis of the Electrospun Scaffolds

The mucoadhesive potential of the developed scaffolds were measured in terms of the adsorption of mucin (porcine stomach type II) by periodic acid/Schiff colorimetric method [19]. Standard mucin solutions at the concentrations of 12.5, 6.25, 3.125, and 1.625 mg per 100 mL of phosphate buffer (pH: 5.5) were prepared, and 200 µL of periodic acid (10%) was added to 2 mL of each sample and incubated for 2 h. After incubation, 200 µL of Schiff reagent were added to the mixtures, and the UV absorbance after 30 min at 555 nm was measured to form the calibration curve. A total of 1 mL of mucin solution (0.125 mg/mL) was added to 1 mL of the filtrate (10 mg of electrospun fibers suspended in 10 mL of phosphate buffer or simulated vaginal fluid) and stirred using a magnetic stirrer for 1 h at 37 °C at 300 rpm. To determine the amount of free mucin, samples were centrifuged at 15,000 rpm for 45 min at 20 °C. Next, 200 µL of periodic acid was added to the supernatants and samples incubated at 37 °C for 2 h, and then 200 µL of Schiff agent was added. The absorbance reading after 30 min of incubation was measured at 555 nm using a UV/vis spectrophotometer.
(2)$$\%\;\mathrm{Mucin}\;\mathrm{adsorption}=\frac{\left(\mathrm{total}\;\mathrm{mass}\;\mathrm{of}\;\mathrm{mucin}\;-\;\mathrm{free}\;\mathrm{mucin}\right)}{\mathrm{total}\;\mathrm{mass}\;\mathrm{of}\;\mathrm{mucin}}\;\times \;100$$

### 2.8. In Vitro Release Studies of Metronidazole from the Electrospun Scaffolds

This was carried out using Franz^®^ diffusion cells with an available diffusional cross-sectional area of 0.98 cm^2^. A 0.45 um white girded Millipore^®^ membrane (CAT. NO. HAWG047S6, LOT. NO. F0JB71372C) was used for the studies. A pH 4.5 phosphate buffer was used as the receptor compartment of the Franz^®^ diffusion cell and filled with 30 mL of fresh buffer 10. The receiver chambers were kept at 37 ± 1 °C in a water bath. Permeant concentrations were determined spectrophotometrically at 322 nm. All measurements were performed in triplicate. The cumulative percentage of metronidazole permeating the membrane (% Q) was plotted as a function of time. The drug flux at a steady state (Jss, µg/cm^2^/h) was calculated from the slope of the linear portion of the cumulative amount permeating the membrane per unit area versus time plot [20].

### 2.9. Biological Characterization 

#### 2.9.1. *Lactobacilli* spp. Viability before and after Electrospinning Process 

To determine the number of colony-forming units (CFU) in the prepared probiotic nanofibers, their weighed pieces were completely dissolved in 5 mL of simulated vagina fluid or sterile water. Following that, a 9-step dilution was carried out (1 step = 10-fold dilution), with 1 mL of solution pipetted onto MRS agar plate after each step. Agar plates were stored in a lockable plastic box with activated Microbiology AnaeroGen^®^ (Thermo Scientific, UK), which created an anaerobic environment by adsorption of oxygen. The generated colonies were counted after being stored at 37 °C for 48 h. The amount of CFUs in 1 g of starting solution was calculated at 100%. This was carried out on the emulsion prepared prior to electrospinning, as well as on the electrospun scaffolds at 48 h and 30 days post-electrospinning after storage at 25 °C [21].

#### 2.9.2. Safety Assessment in an Animal Model

This was carried out using 20 female adult healthy naïve rats with weights around 160–170 g, which were purchased from Nomad Farms, Sango Ota, Ogun State, Nigeria. The rats were acclimatized to the new environment for 8 days prior to study commenced and maintained at 29 ± 2 °C with relative humidity (40 ± 3%) in a 12 h light and dark cycle. The National Institutes of Health guide for the care and use of laboratory animals was followed. The experiments followed with the institutional guidelines and were approved in writing by Health Research Ethical Committee CMUL/HREC/12/21/992. This study was carried out using animal research, and we reported the in vivo experiments, according to the ARRIVE guidelines for documenting the study. All rats were hormonally synchronized by subcutaneous injection with 2 mg/kg/body weight of medroxyprogesterone acetate (Depo Provera^®^, Pfizer, NY, USA) 5 days prior to the electrospun scaffold application. To assess irritation, the rats were randomized into 4 groups consisting of 3 rats/group and were administered daily doses of the formulation intravaginally using a sterile stainless-steel feeding needle with a ball-point end. The vagina’s external appearance was observed daily and difficulty in inoculation, vaginal contraction, excessive screams from the animals, spasms, redness, and burning were noted if present. No form of anesthesia was used during the process, and the rats were humanely euthanized by exposure to carbon dioxide gas on day 14. The vaginal and rectal tissues were excised into sterile sample bottles containing 10% formalin solution for histological analysis, and the sections excised were viewed and photographed using a Lecia DM 750 microscope and ICC HD 50 camera. Reports were made using magnifications of ×100 and ×20 [22].

#### 2.9.3. Statistical Analysis

All reported data (in triplicate) were expressed as mean ± standard deviation of experimental values for each value, and the results were reported accordingly.

## 3. Results

### 3.1. Morphological Characterization

The scaffold sizes were in nanometers (Figure 1). The F1 scaffolds showed thick fibers with a uniform distribution and diameters of 1.207 ± 0.391 nm. The F2 scaffolds showed thinner fibers that were closely packed together and had a diameter of 1.029 ± 0.496 nm. The F3 scaffolds showed large formations that were closely packed and uniformly distributed with diameters of 16.351 ± 6.048 nm. The F4 fibers were closely packed with near uniform surfaces and diameters of 0.957 ± 0.332 nm. The F5 fibers have bead formations and were closely packed with near uniform surfaces and diameters of 0.933 ± 0.304 nm. The F6 fibers were densely packed with bead formations and diameters of 0.921 ± 0.255 nm. The porosity of each scaffold was determined, as follows: F1 had 32.3 ± 2.5%, F2 had 33.4 ± 3.6%, F3 showed the highest porosity of 37.2 ± 2.0%, F4 had 27.5 ± 2.0%, F5 had 31.7 ± 4.9%, and F6 had 20.3 ± 5.5%, which was the lowest porosity. The addition of the surfactant appeared to affect the porosity with a high concentration of the tenside, facilitating lower porosity in the electrospun fibers.

### 3.2. Thermal Analysis of Nanofibers

Thermogravimetric analysis was used to assess the thermal stability of the nanofibers. F1 underwent thermal decomposition in 4 steps, with the first occurring between 300 and 368 °C. F2 and F4 showed 2 weight-loss steps. The metronidazole-loaded formulations F5 and F6 showed a single weight-loss step from 295 to 466 °C, as shown in Figure 2A,B. The electrospun formulations containing *Lactobacillus* spp. (F7 and F8) showed very similar trends with a single weight-loss step observed from around 300 to 350 °C. The formulations displayed thermal stability up to 300 °C when analyzed via differential thermal analysis. The F1, which contained a polymer blend, had a weight loss of 1%, F2 had a weight loss of 4%, F3 had 8%, F4 had 7%, F5 had 10%, F6 had 4%, and finally, F7 and F8 had similar weight losses (Figure 2B).

### 3.3. Chemical Characterization

Fourier transform infrared spectroscopy (FTIR) was employed to determine the interactions between the drugs and polymers. PVA was observed to have characteristic peaks at 3273, 2948, 1234, 1088, and 831 cm^−1^. PCL was observed to have characteristic peaks at 3273, 2948, 1729, 1360, and 115 cm^−1^. For F1, the characteristic peaks were observed at 293, 1726, and 1234 cm^−1^. F2, F3, and F4 showed characteristic peaks common to the polymers (Figure 2C). There was an increasing level of intensity of the 1729 cm^−1^ peak when moving from F2 to F4. The presence of metronidazole in F5 and F6 was confirmed by the absorbance peaks at 1532, 1367, 1256, 1166, 831, and 734 cm^−1^. The FTIR spectra of the electrospun formulations containing *Lactobacillus* spp. (F7 and F8) were very similar to those of the formulations F5 and F6. There were observable peaks in the range from 900 to 1300 cm^−1^. The peak observed around 1750 cm^−1^ was ascribed to the carboxylic acid groups present in the cell walls of *Lactobacillus* spp.

X-ray diffraction (XRD) showed that F1 containing PVA and PCL was significantly amorphous in nature, with characteristic peak at 2 Ɵ-values of 21.8° and 23.8°. These characteristic peaks were not observed in F3 and F4. The characteristic peaks of metronidazole were not observed in the XRD patterns of the metronidazole-loaded formulations of F5 and F6. The differential scanning calorimetry showed sharp endothermic peaks at close to 90 °C for both formulations (F7 and F8). A second peak was observed around 150 °C for F8. F7 was also observed to have that same peak, though it was less intense and had shifted to a higher temperature range, from 150 to 170 °C. The diffused peaks from 200 to 250 °C were also observed (Figure 2D).

### 3.4. Mechanical Characterization

The ultimate tensile strength of the electrospun scaffolds was determined with F1, which contained only the polymer blend of PVA/PCL with the highest UTS of 0.3340 MPa. F2 contained the polymer blend and 0.5% surfactant, and it had a UTS of 0.2015 MPa. F3, with a lower concentration of the surfactant at 0.25%, had a UTS of 0.3038 MPa. F4, which contained the polymer blend and a lower concentration of the surfactant at 0.1% had a UTS of 0.2253 MPa. F5, which contained the polymer blend and metronidazole at 0.5%, had a UTS of 0.1965 MPa. F6, which contained the polymer blend, 0.5% of the surfactant, and metronidazole at 0.5%, had a UTS of 0.1022 MPa (Figure 3A). F7, which contained the polymer blend, metronidazole, and a concentration of the surfactant at 0.25%, had the lowest UTS at 0.0460 MPa. F8, which contained the polymer blend, metronidazole at 0.5%, and a concentration of the surfactant at 0.1%, had a UTS of 0.1079 MPa. The ductility of the electrospun scaffolds was determined, resulting in F7 (29.25 ± 0.83%) and F8 (28.39 ± 0.51%) having the lowest ductility. The tensile strain and stress of the fibers at breaks were the following, respectively: F2 at 1.150 mm/mm and 23.013 MPa, which was the highest; F1 at 1.053 mm/mm and 21.066 MPa; F3 at 0.755 mm/mm and 15.109 MPa; followed by F6 at 0.617 mm/mm and 12.343 MPa; F5 at 0.501 mm/mm and 10.022 MPa; F4 at 0.358 mm/mm and 7.167 MPa; F7 at 0.293 mm/mm and 5.679 MPa; and finally, F8 had the lowest values of tensile strain and stress at breaks in the fibers at 0.284 mm/mm and 5.679 MPa (Figure 3C).

### 3.5. Mucoadhesion Analysis and In Vitro Release Studies of Metronidazole from the Electrospun Scaffolds 

The scaffolds exhibited mucoadhesion from 31.54 ± 0.83% to 57.86 ± 0.95%, where the increased sodium cocoamphoacetate concentration led to increased mucoadhesion. F6 showed the highest mucoadhesion at 57.86 ± 0.95%, as compared to 42.67 ± 1.22% and 50.89 ± 1.01% for the F8 and F7 scaffolds, respectively. The polymers blend, PVA/PCL, had mucoadhesion of 31.54 ± 0.83% which also reflected their inherent mucoadhesive property. Incorporation of the tenside has shown to increase the mucoadhesive property of the scaffold. This inferred that within the vagina, the formulation could adhere to tissue with minimal clearance. The increased mucoadhesion also extended the contact of the formulation with the vaginal tissue. The mucoadhesion was further facilitated via the PCL degradation, where it was first hydrolyzed to hydroxycaproic acid and adipic acid. In addition, this facilitated the attachment to tissues due to the stretching vibrations of the C-H, C=O, epoxy C-O, and alkoxy C-O groups associated with PCL.

The in vitro release was carried out to determine how the active pharmaceutical ingredients were released from the drug formulation and made available at the site of action. The formulated dosages being evaluated were subjected to conditions that could facilitate drug release. The Franz diffusion cells were used, and based on the cumulative amount of metronidazole released, F6 showed an increased release profile, and it contained the highest concentration of the surfactant sodium cocoamphoacetate. We inferred that the presence of the surfactant had stabilized the scaffold formulation and enabled the sustained release of metronidazole over a period of 6 h. The values ranged from 14.8502 µg/cm^2^ at 5 min to 514.937 µg/cm^2^ at 6 h. F7, which had a surfactant decrease from 0.5% to 0.25%, showed a decrease in the cumulative amount of metronidazole released, with values ranging from 2.0191 µg/cm^2^ at 5 min to 109.0317 µg/cm^2^ at 6 h. The F8 formulation released 17.65089 at 5 min to 255.9705 µg/cm^2^ at 6 h. The pattern of release favored a sustained release, reducing the dosing frequency and ensuring patient compliance. The F5 formulation did not contain any surfactant and had a release range from 1.889 µg/cm^2^ at 5 min to 89.4917 µg/cm^2^ at 6 h. The release data were then input to varying kinetic models to determine the mechanism of the drug release, which was observed to follow the Higuchi model (Table 2).

### 3.6. Biological Characterization

#### 3.6.1. *Lactobacilli* spp. Viability before and after Electrospinning Process

The determination of the viability of the *Lactobacillus* spp. incorporated within the electrospun scaffold was critical for ensuring that the bacteria would be available at the site of action upon application. *Lactobacilli* spp. was responsible for the lactic acid formation within the vagina; hence, an adequate amount of colony-forming units had to be viable post-electrospinning, as well as post-storage at 25 ± 0.5 °C. F6, F7, and F8 were loaded with 1 mL of 150 × 10^6^/mL of *Lactobacilli* spp. The emulsion prior to electrospinning exhibited an average of 203, 198, and 214 CFU/mL for F6, F7, and F8, respectively. Furthermore, 48 h post-electrospinning cultures showed 209, 183, and 190 CFU/mL per mg of electrospun fiber *Lactobacillus* spp. for F6, F7, and F8, respectively. At 30-days post-electrospinning, the cultures showed 150 and 143 CFU/mL *Lactobacillus* spp. in F7 and F8, respectively.

#### 3.6.2. Safety Assessment in an Animal Model

As shown in Figure 4A,B, the histological section of the vaginal tissue of the control group showed the normal architecture of vaginal tissue, with the lamina propria (LP) containing blood vessels (BVs) and lymphocytes (LMs) with non-cornified stratified squamous epithelium (EP). Figure 4B,C shows tissues treated with F6, which had normal architecture with lamina propria (LP) containing blood vessels (BVs) and lymphocytes (LMs) with non-cornified stratified squamous epithelium (EP). The adventitia (AD) blood vessels (BVs) as well as the lamina propria (LP) containing blood vessels (BVs) and lymphocytes (LMs) with cornified and hypertrophied stratified squamous epithelium (EP) were also observed. Figure 4D,E shows tissues treated with F7, and these were a slightly abnormal epithelium with distortions on the stratum corneum layer, as indicated by the arrow. The lamina propria (LP) containing blood vessels (BVs) and hypertrophied epithelium (EP) showed estrogen-induced stratification and cornification. The histological slides showed a moderate atrophic lamina propria (LP) and a very thin cornification of the stratum with distortions and a hypertrophied epithelium (EP), as well as estrogen-induced stratification and cornification by day 14. A normal lamina propria (LP) containing blood vessels (BVs) and lymphocytes (LMs) with a hypertrophied epithelium (EP) showed estrogen-induced stratification and cornification at 7 days. F8-treated animals showed normal vaginal architecture with a lamina propria (LP) containing blood vessels (BVs) with a non-cornified stratified squamous epithelium (EP) with a permeable layer. At 14 days, the tissues showed lamina propria (LP) containing blood vessels (BVs) and lymphocytes (LMs) with non-cornified, stratified squamous epithelium (EP).

## 4. Discussion

The inter-individual variability after treatment with metronidazole was observed as high in the general female population due to the response of microbiota to the presence of the antibiotic. BV is the most prevalent form of vaginitis, showing a high prevalence ranging from 23% to 29% across regions [2,3,4]. BV has been linked with an increased risk of contracting sexually transmitted infections (STIs), human immunodeficiency virus (HIV), herpes simplex virus, chlamydia, and gonorrhea [5,6]. In 2019, BV was the most prevalent vaginal infection and was estimated to affect from 5% to 70% of people globally [1]. Vaginal biofilms play a major role in treatment failure, as they possess structures that prevent antimicrobial penetration and the further contact with microorganisms, leading to antibiotics resistance and affecting drug release, which modified drug solubility and reduced drug effectiveness [2,3,4,5,6,7]. Maintaining the diversity and colonization of *Lactobacillus* spp. is important for ensuring complete bacterial clearance, which underlies the need to develop a formulation that is mucoadhesive and contains a tenside for biofilm clearance, an antibiotic, and *Lactobacilli* spp. Electrospun scaffolds that incorporate polymers, metronidazole, and *Lactobacilli* spp. have shown various advantages over other delivery systems for the treatment of BV, such as cost-effectiveness, ease of operation, and controlled release. The latter increases the contact time of the drug at the site of action (the vagina), allowing it to penetrate the biofilm and leading to longer residence times in the vagina in order to prevent recurrence and ensure better therapeutic outcomes. The nanofiber formulation comprised the polymers PVA and PCL due to their associated biodegradability and biocompatibility [23,24]. PVA was chosen for its lubrication properties that enhance the mucoadhesiveness of the nanofiber on the walls of the vaginal tissue, prolonging the contact time and improving therapeutic outcomes [25]. PVA was discovered to enhance the incorporation of bacteria, such as *Lactobacilli,* as was used in the study. Furthermore, PCL and PVA were found to enable sustained release, hence their use in our study [26]. A study reported that the amphoteric tenside sodium cocoamphoacetate was found to be highly effective in the dissolution and inhibition of biofilms. After 20 h, it prevented the formation of the G. vaginalis biofilm completely. There was more than 50% dissolution of the established biofilm and a 60% inhibition of the biofilm viability after 40 h, indicating sodium cocoamphoacetate was the most potent biofilm dissolver in the study [13].

### 4.1. Scanning Electron Microscopy

The microscopic examination of the nanofibers using a scanning electron microscope showed that the PVA decreased the surface tension between the immiscible phases and promoted the formation of more uniform nanofibers by acting as an emulsifying agent. The fiber diameter decreased between formulations, from F1 to F6, except for F3, which had a diameter of 16.351 nm. Inadequate stretching and increased velocity was associated with the large diameter observed in F3, in addition to the reduced surface tension caused by the lower concentration of amphoteric tenside sodium cocoamphoacetate in F3, from 0.5% to 0.25%, which, in turn, had led to the larger fiber diameter and could be considered a contributing factor [27]. The lowest diameter was found in F6 at 0.933 nm. The reduction in the fiber diameter could be associated with the addition of the surfactant, which had stabilized the solution, lowered the surface tension, and boosted the net charge density, which, in turn, increased the uniformity and decreased the diameter of the electrospun nanofibers [27]. The observed beading could be associated with the surface tension, which attempted to reduce the surface area per unit mass [28]. The surfactant concentration influenced the large beaded structure, and Zheng et al. [27] showed previously that larger surfactant concentrations had resulted in less beaded structures, indicating that the larger the surface concentration of sodium cocoamphoacetate, the less beading observed. The presence of sodium cocoamphoacetate reduced the surface tension.

### 4.2. Porosity Determination of the Electrospun Scaffolds

Based on the porosity of the electrospun scaffolds, as shown in Figure 2, the highest porosity observed was in F3 with 37.2 ± 2.0%, followed by F2 with 33.4 ± 3.6%. Both formulations had the addition of the surfactant sodium cocoamphoacetate at varying concentrations. F1, with the polymer matrix, had a porosity of 32.2 ± 2.5%, and F5, with the polymer blend and metronidazole, was next in rank, with 31.7 ± 4.9%. F6 had the lowest porosity at 20.3 ± 5.5%, which strongly indicated that loading different substances at varying concentrations had influenced the porosity of the electrospun scaffolds [15].

### 4.3. Tensile Strength and Ductility of the Electrospun Scaffolds

The highest tensile strength, as shown in Figure 3A, was observed in F1, which contained only the polymer blend, with an ultimate tensile strength of 0.3340 MPa. This was associated with favorable wettability and adhesion between the electrospun scaffold and the matrix at varying concentrations in F1 [15]. F3 had the second best UTS at 0.3038 MPa, whereas F4 had 0.2253 MPa, F2 had 0.2015 MPa, F5 had 0.1965 MPa, F6 had 0.1079 MPa, F7 had 0.1022 MPa, and finally, F8 had 0.0460 MPa. With the increased electrospun-scaffold loading and the decreased concentration of the surfactant, the tensile strength was observed to decrease; therefore, these factors played an important role in determining the integrity of the electrospun scaffold formulations. The scaffold ductility was observed to decrease with the increased loading. F7 and F8 had the lowest ductility values of 29.25% and 28.39%, respectively, which was attributed to the clustering of particles that prevented the mobility of crazes. The highest ductility, as shown in Figure 3B, was observed in F2, which exhibited the best ability to deform without breaking with a ductility of 93.83%. This was attributed to the addition of the surfactant that increased the affinity of the components, resulting in high ductility [29]. The surfactant increased the miscibility and adhesive forces between the electrospun scaffolds, resulting in a positive outcome. PVA was shown to have good deformation and flexibility [29]. F8 exhibited the least ductility at 28.39% with a lower concentration of *Lactobacillus* spp. The clustering of the particles, which resulted in each component exhibiting a rigid behavior, prevented the crazes from moving freely during the tensile loading and could have been the cause of the poor ductility [15].

### 4.4. FTIR Spectroscopy 

FTIR spectroscopy was employed to elucidate the interactions between the drug and the polymer, as shown in Figure 2C. The figure shows the FTIR spectra of the individual polymers (PVA, PCL, and CAP), as well as the various formulations of the drug delivery system (F1–F6). The pure PVA had characteristic peaks at 3273, 2948, 1703, 1234, 1088, and 831 cm^−1^. The peak at 3273 cm^−1^ was attributed to the O–H stretching of the hydroxyl groups as a result of the intramolecular hydrogen bonding [30]. The asymmetric CH2 stretching vibration and carbonyl (C=O) stretching were attributed to the peaks at 2948 and 1703 cm^−1^, respectively, while the bending vibration of the C-H bond in the CH2 group, the C-O stretching in the acetyl groups, and the C-C vibrational stretching were attributed to the peaks at 1234, 1088, and 831 cm^−1^, respectively. Similar spectra for pure PVA had been reported previously [31,32]. The FTIR spectra of pure PCL are also shown in Figure 2C. The characteristic peaks were observed at 2948, 1729, 1360, and 1155 cm^−1^, and these were assigned to the stretching vibrations of the C-H, C=O, epoxy C-O, and alkoxy C-O groups. These results agreed with those previously reported by Rezaei and Mohammadi in 2013 and by Trakoolwannachai et al. [33,34]. The pure CAP shared similar spectral characteristics with PCL. For the different formulations, the spectra of F1 (which contained only PVA and PCL) were similar to those of PCL. This could be linked to the use of more PCL, as compared to PVA, in the formulation. The major peaks were recorded at 2937, 1726, and 1234 cm^−1^, which were attributable to the C-H stretching in the CH2 group, the C=O stretching of the carbonyl group in the PCL, and the C-O in PVA. These findings were similar to those reported by Maheshwari et al. and Rajeswari et al. [35,36]. The F2, F3, and F4 formulations contained PVA, PCL, and CAP, and their spectra showed the characteristic peaks common to these 3 compounds. However, the prominent peak at 1725 cm^−1^ for PCL shifted slightly to 1729 cm^−1^ in the F2, F3, and F4 formulations. There was also an increasing level of intensity in the 1729 cm^−1^ peak between F2 and F4. This could be attributed to the reduction in the level of CAP (0.5 in F2 to 0.1 in F4) in the formulation, thus giving significance to the level of PCL. The F5 and F6 formulations contained the drug (metronidazole), and they showed similar spectral trends. However, the presence of metronidazole in these formulations was confirmed by the absorbance peaks at 1532 cm^−1^ (N-O stretching), 1367 cm^−1^ (symmetric NO2 stretching), 1256 cm^−1^ (OH deformation), 1166 cm^−1^ (C-O stretching), 831 cm^−1^ (C-N stretching), and 734 cm^−1^ (CH2 vibration). Similar observations had also been reported by Celebioglu and Uyar [37]. The FTIR spectra of the electrospun formulations containing *Lactobacillus* spp. (F7 and F8) were very similar to those of the F5 and F6 formulations. There were observable peaks from 900 to 1300 cm^−1^, which were attributed to the nucleic acids and the proteins in *Lactobacillus* spp. The authors of [38,39,40] had reported similar peak assignments observed in *Lactobacillus*-loaded electrospun scaffolds consisting of sodium alginate and poly (vinyl alcohol). The peak observed around 1750 cm^−1^ was attributed to the carboxylic acid groups present in the cell walls of *Lactobacillus* spp. Similar observations had been reported by Amiri et al. [41] in *Lactobacillus acidophilus*-loaded lactose/whey-protein nanocomposite. Other studies had also reported these peaks in for *Lactobacillus*-loaded electrospun scaffolds, which the authors had attributed to the overlap of the polymer peaks that confirmed the encapsulation of *Lactobacillus* spp. [42,43].

### 4.5. XRD Analysis

An XRD analysis was used to elucidate the physical state (either crystalline or amorphous) of the drug (metronidazole) in the different formulations, and the results are shown in Figure 2D. The F1 formulation containing PVA and PCL was significantly amorphous in nature, according to the characteristic peak at the 2θ values of 19.5° and 20.1°, which was indicative of the presence of PVA [44]. PCL conferred a slight crystallinity to the formulation, as observed in the peaks with 2θ values of 21.8° and 23.8°, which were characteristic peaks belonging to the orthorhombic planes of PCL [45]. According to the literature, the XRD pattern of the metronidazole would show distinctive diffraction peaks at the 2θ values of 13°, 25°, 28°, 30°, and 34° because of its crystalline nature [46]. However, the characteristic peaks of metronidazole were not observed in the XRD pattern of the metronidazole-loaded formulations (F5 and F6), suggesting that metronidazole had lost its crystalline nature after being encapsulated in the amorphous polymer matrix. This observation had been reported in previous research [47,48]. This was, however, not a concern, as noted by Celebioglu and Uyar [37], who had reported that the existence of metronidazole in the amorphous state in the drug formulation was desirable. The amorphous state of the metronidazole in the formulation had enabled the fast dissolution of the drug, which then facilitated the delivery process [21,49].

### 4.6. Thermal Analysis of the Electrospun Scaffolds

The thermal stability of the formulations was assessed using TGA and DTA. Their respective curves, as shown in Figure 2A,B, indicated the percentage of weight loss as a function of temperature. The results showed the F1 formulation containing PVA and PCL underwent thermal decomposition in 4 steps. The first weight-loss step occurred between 300 and 368 °C, and this was attributed to the loss of the water molecules immobilized within the polymer matrix [50]. The TGA profiles for F2 and F4 showed 2 weight-loss steps. The second weight-loss step was attributed to the removal of the water chemically bound to the functional groups in PVA and PCL, which were not removed during the first step [48]. The metronidazole-loaded formulations (F5 and F6) showed a single weight-loss step, from 295 to 466 °C, which was possibly due to the degradation of the metronidazole [51]. These profiles were corroborated by the derivative curves (DTA profiles) shown in Figure 4. The formulations displayed good thermal stability up to 300 °C. From 0 to 300 °C, F1 only recorded a weight loss of about 1%. F5, which contained metronidazole, recorded a 10% weight reduction within the same temperature range. The additional weight loss could be attributed to the presence of metronidazole [48]. In contrast to F6, which also contained metronidazole, the weight loss recorded within the same temperature range was only 4%. The lower weight loss could be attributed to the addition of sodium cocoamphoacetate to the F6 formulation, which could have enhanced its thermal stability. When comparing the profiles of F2, F3 and F4, weight losses of 4%, 8%, and 7% were observed. The lower weight losses, as compared to F5, were also attributed to the addition of sodium cocoamphoacetate in these formulations. The TGA and DTA profiles of the electrospun formulations containing *Lactobacillus* spp. (F7 and F8) showed very similar trends with a single weight-loss step observed from around 300 to 350 °C.

### 4.7. DSC Thermogram of the Formulations

The DSC thermogram for the formulations containing *Lactobacillus* spp. (F7 and F8) showed a sharp endothermic peak close to 90 °C for both formulations and was attributed to the glass transition of PVA [52]. A second peak was observed around 150 °C for F8. Although this peak was also represented in F7, it was, however, less intense and at a higher temperature range of 150–170 °C. This could have been caused by the higher concentration of sodium cocoamphoacetate in F7, which increased the glass transition [53]. The diffused peaks around 200–250 °C were attributed to the melting temperature of PVA [54].

### 4.8. Mucoadhesion Analysis and In Vitro Release Studies of Metronidazole in the Electrospun Scaffolds

The mucoadhesion analysis of the scaffolds showed a mucoadhesion between 31.54 ± 0.83% and 57.86 ± 0.95%, where the increased sodium cocoamphoacetate concentration had led to increased mucoadhesion. F6 showed the highest mucoadhesion, as compared to the F7 and F8 scaffolds. The polymers blend, PVA/PCL, had a mucoadhesion of 31.54 ± 0.83%, which also reflected their inherent mucoadhesive properties. This was essential for increased drug release and retention along the vaginal mucosa, which would improve the contact time and ensure complete clearance of bacteria and vaginal biofilms [19].

When the Higuchi model was used to analyze the drug release, the electrospun scaffolds had high r^2^ values of 0.954–0.9933, and it was observed that the release followed the Higuchi diffusion kinetics, according to the fitted data. The Higuchi cumulative plot was expressed as the percentage of cumulative metronidazole release versus the square root of time; therefore, we could deduce that the amount of drug release was proportional to the square root of time. Furthermore, within the delivery system, diffusion caused the drug release [55]. The Higuchi square root of time model was developed from Fick’s first law of diffusion, and its applicability suggested that the homogeneous planar matrix, which did not disintegrate, was where the medication would be released. In these circumstances, the medication slowly diffused from the polymer matrix over time [56]. To understand the drug-release mechanism, the release of metronidazole from the polymer matrix was fitted into the Korsemeyer–Peppas equation, and the n-values obtained were 0.8334, 0.9664, 0.8838, and 0.8181 for F5, F6, F7, and F8, respectively. Due to the drug’s erratic diffusion, their n-values indicated a non-Fickian diffusion-release mechanism, which was a swelling-and-diffusion mechanism. In general, the anomalous transport pointed to a drug-discharge mechanism that combined both diffusion and erosion as 0.5 ≤ n ≤ 1.0 [55].

### 4.9. Safety Assessment in an Animal Model

Safety assessments were carried out using the naïve female rats, and no electrospun scaffolds, except for F7, displayed irritation or inflammation in the vaginal epithelium of the rats, as compared to the vaginal epithelium of control (no treatment) group. The histological section showed normal lamina propria with blood vessels and lymphocytes, with the exception of those treated with F7, as they showed a moderately atrophic lamina propria. F6 had a protection layer against invading pathogens and against water loss in the stratified squamous epithelium. F7 showed cornification and stratification, indicating an impermeable layer of flattened cells but with moderate atrophy. The short-term use of the proposed formulation was determined to be safe. If 14-day use was necessary, a lower concentration of the tenside would be required, such as in F8. All the formulations were deemed to be safe without any adverse effects on vaginal tissue; however, only F8 exhibited an absence of atrophy and inflammation in the vaginal epithelium of the rats, as compared to the control group with no treatment.

## 5. Conclusions

This study reported the successful development of long-lasting electrospun scaffolds for the management of bacterial vaginosis by the incorporation of a tenside, metronidazole, and *Lactobacilli* spp. The polymer formulation incorporated polyvinyl alcohol and polycaprolactone. The electrospun scaffold F8 had adequate porosity that would enable vaginal secretion to infiltrate the scaffold matrices. The electrospun scaffolds had adequate tensile strength, and the release was facilitated by the addition of sodium cocoamphoacetate for the sustained release. The release followed the Higuchi model and showed a non-Fickian diffusion-release mechanism, where the anomalous transport pointed to a drug-discharge mechanism that combined both diffusion and erosion. The developed electrospun scaffold F8 was stable and generally safe on vaginal tissues and maintained acidic pH, which was effective for the management of bacterial vaginosis, and is recommended for additional research in the future.

## Figures and Tables

**Figure 1 pharmaceutics-15-01263-f001:**
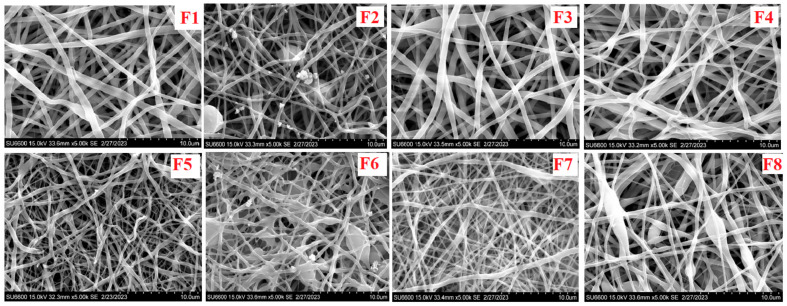
Scanning electron micrographs of developed scaffold formulations **F1**–**F8**.

**Figure 2 pharmaceutics-15-01263-f002:**
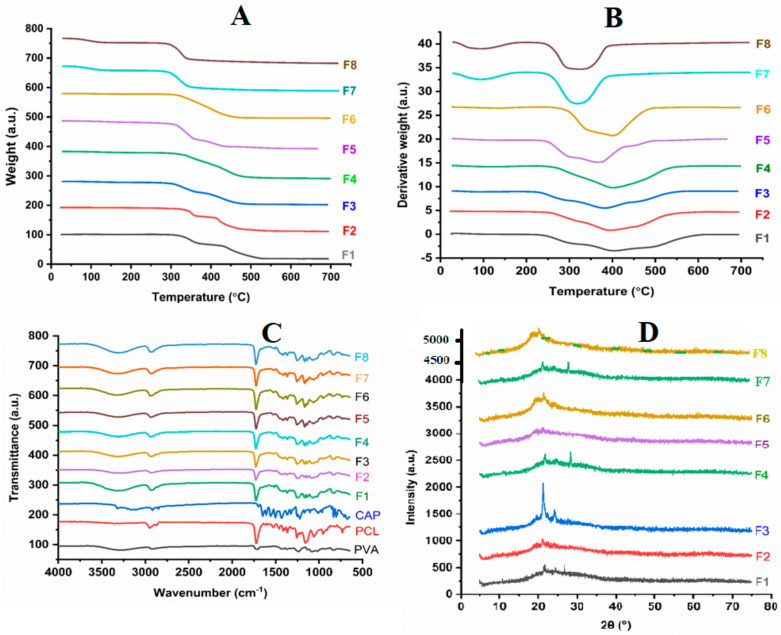
Morphological characterization of the electrospun scaffolds using (**A**) TGA, (**B**) DTA, (**C**) FTIR, and (**D**) XRD.

**Figure 3 pharmaceutics-15-01263-f003:**
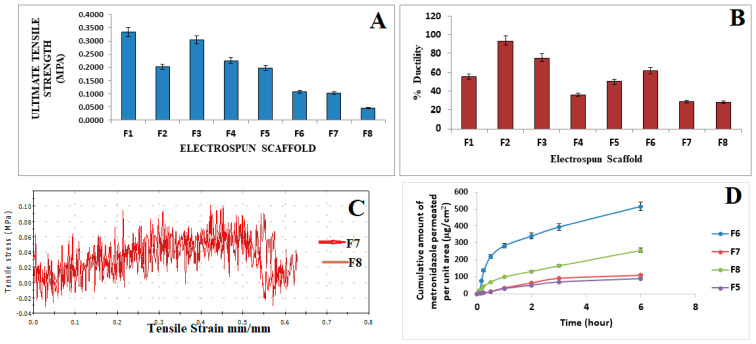
(**A**) Ultimate tensile strength of electrospun scaffolds, (**B**) Ductility of electrospun scaffolds, (**C**) Tensile strain vs. tensile stress of electrospun fibers F7 and F8, and (**D**) Cumulative Amount of metronidazole permeated per unit (µg/cm^2^) vs. time in hours.

**Figure 4 pharmaceutics-15-01263-f004:**
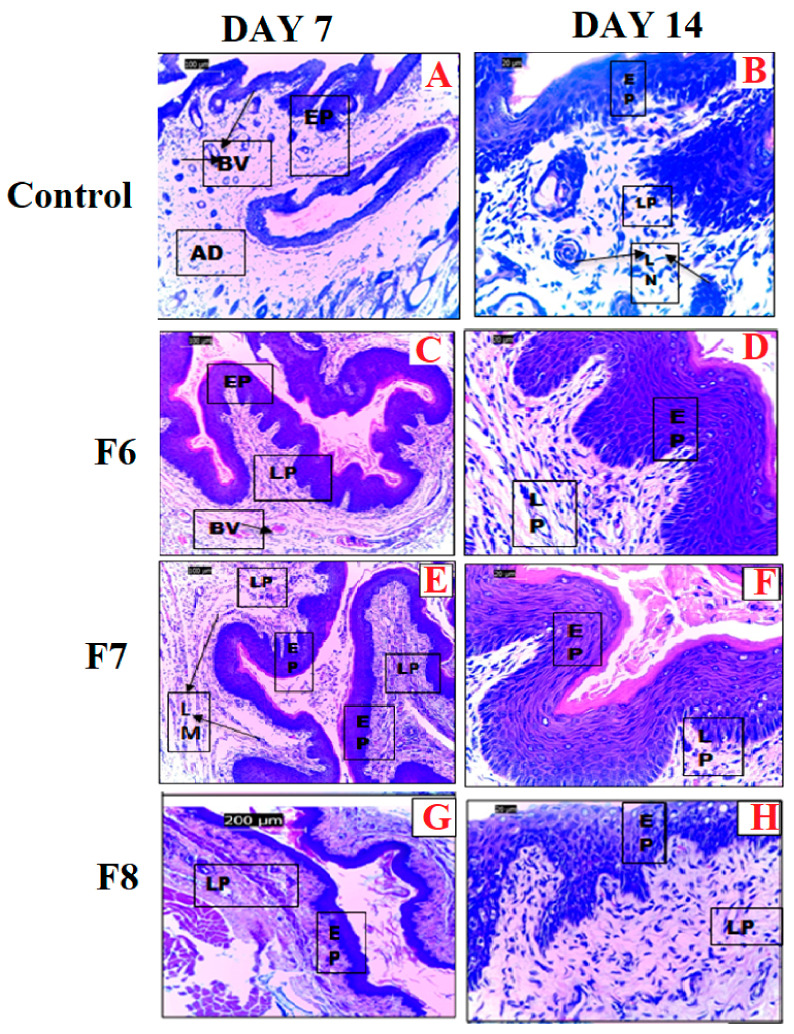
Photomicrographs (H & E ×100) of the histology slides of a section of the vaginal tissue treated with (**A**,**B**) control/no treatment, (**C**,**D**) electrospun scaffolds F6, (**E**,**F**) electrospun scaffolds F7, and (**G**,**H**) electrospun scaffolds F8. Lamina propria (LP), blood vessels (BVs), lymphocytes (LMs), squamous epithelium (EP), adventitia (AD).

**Table 1 pharmaceutics-15-01263-t001:** Formulation design for electrospinning solutions PVA/PCL.

Formulation	Metronidazole (%*w*/*v*)	Sodium Cocoamphoacetate (CAP) (%*v*/*v*)	*Lactobacilli* spp./mL
F1	-	-	-
F2	-	0.5	-
F3	-	0.25	-
F4	-	0.1	-
F5	0.5	-	-
F6	0.5	0.5	-
F7	0.5	0.25	150 × 10^6^
F8	0.5	0.1	150 × 10^6^

**Table 2 pharmaceutics-15-01263-t002:** In vitro drug-release kinetic study for metronidazole.

	Zero Order		First Order		Higuchi		Korsmeyer–Peppas	
Formulation	r^2^	k_0_	r^2^	k_1_	r^2^	k_2_	r^2^	n
F5	0.3757	0.0483	0.6459	0.0047	0.9701	4.19	0.9605	0.8334
F6	0.4861	0.1616	0.3335	0.0041	0.954	22.244	0.8618	0.9664
F7	0.563	0.0448	0.6531	0.0050	0.9549	5.2585	0.9535	0.8838
F8	0.8942	0.1616	0.4405	0.0039	0.9933	10.554	0.9213	0.8181

## Data Availability

Not applicable.

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
