# Peer review of "Development of Mucoadhesive Electrospun Scaffolds for Intravaginal Delivery of Lactobacilli spp., a Tenside, and Metronidazole for the Management of Bacterial Vaginosis"

_pharmaceutics, 2023, doi:10.3390/pharmaceutics15041263_

Round 1

Reviewer 1 Report

Reviewer’s comments:

The revised manuscript entitled ‘Development of mucoadhesive electrospun scaffolds for intravaginal delivery of Lactobacilli spp, a tenside and Metronidazole for management of bacterial vaginosis’ has been peer-reviewed. The authors have fabricated novel mucoadhesive polyvinyl alcohol and polycaprolactone electrospun nanofibrous scaffolds for vaginal delivery incorporating metronidazole, a tenside, and Lactobacilli spp. The manuscript can be accepted for publication after addressing the following queries.

Minor concerns

1) The scaffolds exhibited mucoadhesion of between 31.54 ± 0.83 % to 57.86 ± 0.95%, where increased sodium cocoamphoacetate concentration led to increased mucoadhesion. F6 showed the highest mucoadhesion with 57.86 ± 0.95%, compared to 42.67 ± 1.22%, and 50.89 ± 1.01% for F8 and F7 scaffolds respectively. The polymers blend, PVA/PCL had mucoadhesion of 31.54 ± 0.83 % which also reflected their inherent mucoadhesive property. (Page 9)

The authors should clarify the reasons behind the mucoadhesive property of the developed drug system mentioning the functional groups and the type of bondage.

2) Figure 4. Expansion of EP, BV, AD, etc., can be provided in the figure legend.

3) Units and molecular formulas can be placed properly.

a) Backscattering images were acquired under variable pressure mode, at a 60 Pa vacuum with N2, and a 15 kV acceleration voltage. (N2)

b) The samples were then heated in platinum pans from 25 oC to 200oC at the heating rate 10oC/min under

nitrogen atmosphere [10]. (°C)

c) To characterize, the absorption bands of the scaffolds, spectra were recorded in the region of wave numbers between 500 and 4000 cm-1. (cm−1)

Author Response

Point 1: 1) The scaffolds exhibited mucoadhesion of between 31.54 ± 0.83 % to 57.86 ± 0.95%, where increased sodium cocoamphoacetate concentration led to increased mucoadhesion. F6 showed the highest mucoadhesion with 57.86 ± 0.95%, compared to 42.67 ± 1.22%, and 50.89 ± 1.01% for F8 and F7 scaffolds respectively. The polymers blend, PVA/PCL had mucoadhesion of 31.54 ± 0.83 % which also reflected their inherent mucoadhesive property. (Page 9)

The authors should clarify the reasons behind the mucoadhesive property of the developed drug system mentioning the functional groups and the type of bondage

Response 1: Incorporation of the tenside has shown to increase the mucoadhesive property of the scaffold. This infers that within the vagina the formulation will adher to tissue with minimal clearance. The increase mucoadhesion willl also facilitate extended contact of the formulation with the vagina tissue hence facilitiating. Mucoadhesion is further facilitated via PCL degradation where it first hydrolyzes to hydroxy capronic acid and to adipic acid. This facilitates attachment to tissues also due to the stretching vibrations attributable to the C-H, C=O, epoxy C-O and alkoxy C-O groups associated with PCL.

Point 2: Figure 4. Expansion of EP, BV, AD, etc., can be provided in the figure legend

Response 2: This has been provided in the legend. The legend now reads

Figure 4. Photomicrographs (H & E ×100) of the histology slides produce from a section of the vagina tissue treated with (A, B) control/no treatment (C, D) electrospun scaffolds F6  (E, F) elec-trospun scaffolds F7 (G, H) electrospun scaffolds F8. [lamina propria (LP), blood vessel (BV), lymphocytes (LM), squamous epithelium (EP), Adventitia (AD).

Point 3: 3) Units and molecular formulas can be placed properly.

  1. a) Backscattering images were acquired under variable pressure mode, at a 60 Pa vacuum with N2, and a 15 kV acceleration voltage. (N2)
  2. b) The samples were then heated in platinum pans from 25 oC to 200oC at the heating rate 10oC/min under

nitrogen atmosphere [10]. (°C) 

  1. c) To characterize, the absorption bands of the scaffolds, spectra were recorded in the region of wave numbers between 500 and 4000 cm-1. (cm−1)

Response 3: This has been corrected through out the manuscript.

Reviewer 2 Report

The paper presents a promising approach and collected considerable data publishable in Pharmaceutics. I must congratulate the authors on the quality of their study. After my reading, I identified a few issues that could be considered in order to complement the document.

1) The title is not informative. It should provide a general overview of the main findings instead of describing the methods;

2) Abstract and introduction are adequate;

3) The authors could provide a representative image of the formulations;

4) Swelling index considering the biological flux of vaginal fluids must be assessed for the formulations;

5) Statistical evaluation is missing and must be included for proper data comprehension and interpretation.

6) If it is semisolid, the rheological profile should be assessed;

7) The safety was assessed by exposing the animals to the formulations, which should be reconsidered for future studies. There are several alternative methods to evaluate biocompatibility. I suggest to the authors include this topic in the discussion section as a potential limitation of their study.

8) Minor typographical errors in the text.

Author Response

Point 1 The title is not informative. It should provide a general overview of the main findings instead of describing the methods.

Response 1: I kindly think the title is descriptive of the research…..depicting a novel delivery form.

2) Abstract and introduction are adequate.

Response 2: This is noted.

3) The authors could provide a representative image of the formulations.

Response 3: This has been provided via Scanning electron micrographs in Figure 1.

4) Swelling index considering the biological flux of vaginal fluids must be assessed for the formulations.

Response 4: This was not necessary because drug release (Flux) was evaluated using Franz diffusion cell studies. The fibers dissolved in simulated dissolution fluid to facilitate drug release.

5) Statistical evaluation is missing and must be included for proper data comprehension and interpretation

Response 5: This has been included to read…... Statistical analysis: All reported data (in triplicate) are expressed as mean ± standard deviation of experimental values for each values and results are reported as such.

6) If it is semisolid, the rheological profile should be assessed.

Response 6: Electrospun fibers are solid dosage forms.

7) The safety was assessed by exposing the animals to the formulations, which should be reconsidered for future studies. There are several alternative methods to evaluate biocompatibility. I suggest to the authors include this topic in the discussion section as a potential limitation of their study.

Response 7: Cell line studies can be used to evaluated biocompatibility; however, utilization of animals is also a viable method used to evaluate biocompatibility hence it is not a limitation of the study.

8) Minor typographical errors in the text.

Response 8: This has been edited throughout the manuscript.

Reviewer 3 Report

The manuscript on the topic “Development of mucoadhesive electrospun scaffolds for intravaginal delivery of Lactobacilli spp, a tenside, and Metronidazole for management of bacterial vaginosis” is interesting and within the scope of the journal. But the manuscript needs major revision for reconsideration.

The comments are mention below for authors kind attention.

1.      Provide the statement describing the one or more key hypotheses that the work described in the manuscript was intended to confirm or refute. Inclusion of a hypothesis statement makes it simple to contrast the hypothesis with the most relevant previous literature and point out what the authors feel is distinct about the current hypothesis (novelty).

2.      Please write your text in good English (American or British usage is accepted, but not a mixture of these). English language of the manuscript may require editing to eliminate possible grammatical or spelling errors and to conform to correct scientific English.

3.      The abstract should be rewritten. The abstract should state briefly the purpose of the research, the principal results and major conclusions. Numerical values for the most important findings should be reported.

4.      Scientific names are always italicized. Completely revised throughout the script.

5.      Introduction should contain the latest references and discussion related to current topics.

6.      Is the Fabrication of Metronidazole/ sodium cocoamphoacetate loaded nanofibres is author own protocol or adopted from literature?

7.      All the characterization used in the script such as FTIR, XRD, etc there operating conditions, instruments' accuracy preciseness and calibration, and detection limits that need to be added.

8.      Discussion section needs to be more elaborate.

Author Response

Point 1: Provide the statement describing the one or more key hypotheses that the work described in the manuscript was intended to confirm or refute. Inclusion of a hypothesis statement makes it simple to contrast the hypothesis with the most relevant previous literature and point out what the authors feel is distinct about the current hypothesis (novelty).

Response:  Hypothesis 1: a novel mucoadhesive electrospun nanofibrous scaffolds for vaginal delivery can be developed to contain Metronidazole, cocoamphoacetate a tenside and Lactobacilli spp. for management of BV.

Hypothesis 2: An electrospun scaffold drug delivery platform can be used to prevent recurrence of BV via sustained release of metronidazole.

Hypothesis 3: Lactobacilli embedded within the electrospun scaffolds were viable 30 days post electrospinning and the novel formulation was safe in an animal model.

The conclusion answers all the questions raised above. The hypothesis statements don’t however fit into the journal style and is listed here for the reviewer to acknowledge the authors had research questions which were answered.

  1. Please write your text in good English (American or British usage is accepted, but not a mixture of these). English language of the manuscript may require editing to eliminate possible grammatical or spelling errors and to conform to correct scientific English.

Response: This has been done throughout the manuscript

  1. The abstract should be rewritten. The abstract should briefly state the purpose of the research, the principal results and major conclusions. Numerical values for the most important findings should be reported.

Response: The abstract shows purpose of research “The aim of this study is to develop a novel mucoadhesive polyvinyl alcohol and polycaprolactone electrospun nanofibrous scaffolds for vaginal delivery incorporating metronidazole, a tenside and Lactobacilli . This approach to drug delivery seeks to combine an antibiotic for bacterial clearance, a tenside biofilm disruptor, and a lactic acid producer to restore healthy vaginal flora and prevent bacterial vaginosis recurrence.” Numerical values of important findings have been included in the abstract.

  1. Scientific names are always italicized. Completely revised throughout the script.

Response: This has been done throughout the manuscript

  1. Introduction should contain the latest references and discussion related to current topics.

Response:  The introduction contains references

  1. Is the Fabrication of Metronidazole/ sodium cocoamphoacetate loaded nanofibres is author own protocol or adopted from literature?

Response: This is the authors protocol

  1. All the characterization used in the script such as FTIR, XRD, etc there operating conditions, instruments' accuracy preciseness and calibration, and detection limits that need to be added.

Response: Parameters under which the material was tested have been included within the manuscript. All equipment used was calibrated and were deemed accurate before use hence the absence of calibration data within the manuscript.

  1. Discussion section needs to be more elaborate.

Response: This has been carried out.

Round 2

Reviewer 3 Report

The authors have revised the manuscript.